# Insight into Iron, Oxidative Stress and Ferroptosis: Therapy Targets for Approaching Anticancer Strategies

**DOI:** 10.3390/cancers16061220

**Published:** 2024-03-20

**Authors:** Marialuisa Piccolo, Maria Grazia Ferraro, Federica Iazzetti, Rita Santamaria, Carlo Irace

**Affiliations:** BioChem Lab, Department of Pharmacy, School of Medicine and Surgery, University of Naples “Federico II”, 80131 Naples, Italy; marialuisa.piccolo@unina.it (M.P.); federica.iazzetti@unina.it (F.I.); rita.santamaria@unina.it (R.S.)

**Keywords:** iron, oxidative stress, ROS, ferroptosis, cancer

## Abstract

**Simple Summary:**

In this manuscript, we review the progress in knowledge made over the last decade, which, thanks to the commitment of many academics and researchers, has clarified many aspects relating to ferroptosis and its connections with cancer. Ferroptosis is currently regarded as a distinct type of regulated cell death (RCD), characterized by iron-dependent oxidative stress and the accumulation of lethal oxidized lipids. With a focus on the recent literature, the connection between iron homeostasis, oxidative stress and lipid metabolism is highlighted, which overall regulates ferroptotic cellular death. Moreover, special attention is devoted to the possible activation of this RCD pathway as a mechanism of tumor suppression. Its in-depth understanding from a regulatory and molecular perspective could provide important information for the development of new candidate drugs for the treatment of tumors resistant to conventional therapies.

**Abstract:**

Based on the multifaceted molecular machinery that tightly controls iron cellular homeostasis, this review delves into its paradoxical, potentially dangerous role in biological systems, with a special focus on double-edged sword correlations with cancer. Indeed, though iron is a vital micronutrient and a required cofactor participating in several essential cell functions, its tendency to cause oxidative stress can be related both to cancer risk and to the activation of cancer cell death pathways. In this scenario, ferroptosis refers to an iron-dependent form of regulated cell death (RCD) powered by an overload of lethal peroxides sharing distinctive oxidized phospholipid profiles. As a unique cell death pathway, ferroptosis is both morphologically and mechanistically different from other types of programmed cell death involving executioner family proteins. The accumulation of cytotoxic lipid peroxides encompasses a cellular antagonism between ferroptosis execution and defense systems, with iron-dependent death occurring when ferroptosis-promoting activities significantly exceed the cellular antioxidant defenses. The most recent molecular breakthroughs in the execution of ferroptosis have aroused great consideration in tumor biology, as targeting ferroptosis can provide new tools for exploring therapeutic strategies for tumor suppression. Mutations and death/survival pathway alterations, as well as distinctive metabolic regulations of cancer cells, including the propensity to generate ROS, are seen as features that can render cancer cells unprotected to ferroptosis, thereby exposing vulnerabilities which deserve further attention to be regarded as targetable for cancers with limited therapeutic options.

## 1. A Brief Historical Remark about Ferroptosis and Its Main Features

Ferroptosis is a very recently discovered iron-dependent type of cell death. This term was first used in 2012 in the scientific community by Stockwell and co-workers at Columbia University, who also described several of the molecular features of this non-apoptotic regulated cell death (RCD) pathway [1]. Actually, this story began way back in 2003 with the identification of some small anticancer molecules capable of causing non-apoptotic death in specific engineered tumorigenic cells (see Figure 1). Since the RAS family small GTPases are mutated in approximately 30% of cancers, researchers were engaged in the search for selectively lethal compounds to oncogenic RAS mutant cells [2]. Moving in this direction, through a large-scale synthetic lethal chemical screening in engineered human tumorigenic cells, small molecules referred to as RAS-selective lethal (RSL) molecules were identified, including one named erastin, which stands for “eradicator of RAS and ST-expressing cells” [3].

This molecule was effective in killing RAS mutant cells by triggering a non-apoptotic iron-dependent type of cell death marked by oxidative stress that was puzzling to researchers at the time [4,5]. By deepening the mechanism of action, we now know that erastin is capable of starting ferroptotic cell death by activating voltage-dependent anion channels (VDACs) and through the functional inhibition of the cystine–glutamate antiporter system (system X_C_^−^), potentially with possible interactions with additional unknown targets. Thus, cells treated with erastin are depleted of cysteine and unable to synthesize the antioxidant glutathione, leading to excessive lipid peroxidation and subsequent cell death [6]. It became increasingly clear that iron was a central player in the activation of this RCD process, so much so as to support the term ferroptosis, adopted in 2012. Indeed, ferroptosis was avoided by exogenous iron chelation or through the inhibition of cellular iron uptake, confirming the exceptionality of a breakthrough concerning an iron-dependent RCD process [5,7]. 

Conceptually, the accepted sequence underlying ferroptosis activation acts through a multifactorial deregulation of cellular iron homeostasis with increased cytosolic levels of metal ions, which, in turn, trigger oxidative stress coupled with irreversible damage to cellular structures and biomolecules [8]. This cellular dynamic process could have implications for numerous pathophysiological conditions, including cancer and neurodegeneration [9,10]. More recently, in 2018, the Nomenclature Committee on Cell Death (NCCD) established ferroptosis as a unique form of regulated and programmed cell death (PCD) [11]. As a consequence, scholars have forthwith moved countless resources in the search of conceivable causes regarding iron metabolism imbalances that navigate cells towards ferroptosis, thus giving new vigor to the study of the molecular mechanisms underlying deregulations of iron homeostasis, which holds great interest for human health [12]. However, the correlation between iron homeostasis, free cellular iron (the so-called LIP, i.e., labile iron pool, often also referred to as the chelatable iron pool) and oxidative stress has already been known for quite some time, as can be deduced from several scientific papers published at the end of the last century [13,14]. Deregulations of cellular iron homeostasis can translate into significant alterations in the LIP with respect to physiological conditions, frequently associated with morbidity. In particular, LIP increases are related to the possible induction of iron-dependent oxidative stress [15,16]. In these circumstances, excess iron disrupts cellular redox homeostasis, and the production of reactive oxygen species (ROS) can greatly propagate from the Fenton and Haber–Weiss reactions, causing irreversible damage to biomolecules and finally cell death [17,18]. 

Something similar was described in 2001 by Maher and co-workers, who discovered an oxidative-dependent programmed cell death pathway, proposed at the time as “oxytosis” [19]. While they did not describe the connection with iron, to date, there is good experimental evidence for considering oxytosis and ferroptosis as the same cell death pathway [20].

## 2. Correlations between Iron Regulation and Ferroptosis

As the name itself suggests, ferroptosis is intrinsically linked to iron contained in biological systems [21]. Currently, we have definitively assumed that ferroptosis is a definite RCD process and that its occurrence is iron-dependent. Concerning the study of cellular and systemic iron metabolism regulation, in recent years, a substantial share of scientific production has undergone a change in direction, turning distinctively towards the investigation of iron control and ferroptosis connections. Indeed, this trend is confirmed by a growing number of scientific papers (both original research papers and reviews) published since 2012 featuring the word “ferroptosis” in their title (Figure 2). 

As a proof of concept, ferroptosis is mainly characterized by iron excess, the release of reactive oxygen species (ROS), lipid peroxidation and cell damage. The assumption that iron represents an essential mineral trace element ensuring specific biological performances, but at the same time a potential biohazard for living cells, is considered an actual dogma of up-to-date molecular biology [18]. In animal cells, iron can exist in thermodynamically stable forms in both the ferrous (Fe^2+^) and ferric states (Fe^3+^), and its ability to form complexes with a variety of organic molecules has great biological implications. Actually, some iron-binding proteins can also generate high-valent iron intermediates, i.e., Fe^4+^ and Fe^5+^, during their catalytic activity. Living organisms require continuous interconversion between the iron oxidation forms to promote fundamental reactions and maintain redox balances [22]. In particular, the interconversion of Fe^2+^ and Fe^3+^ species facilitates many electron transfer and acid–base reactions necessary in biology. Indeed, iron participates in a number of essential cell functions and, hence, is a vital micronutrient. As such, to form iron–sulfur clusters and iron–oxo centers, or as a central coordinator ion in the heme prosthetic groups, iron works as a required cofactor in metalloproteins, which, in turn, are fundamental for a variety of biochemical processes in aerobic cells and organisms, ranging from cell replication to the transport and utilization of molecular oxygen, thus boosting bioenergetic pathways, including energy extraction from the environment [22]. As remarked, iron accumulation in biological systems can cause toxicity. In this condition, a deregulation of cellular iron homeostasis can trigger significant prooxidative processes with possible lethal consequences for living cells [23]. A plethora of both endogenous and exogenous conditions/factors, as well as inherited or secondary factors, may be the basis of iron overload [24]. In this context, membrane proteins such as the transferrin receptor (TfR) and ferroportin (FPN) can play a key role in controlling iron cellular intake and efflux, respectively, in response to various stimuli. Alterations in their activity and expression can also contribute to iron overloading disorders [18,24]. Then, via ROS production, iron excess can induce multi-organ toxicity through irreversible damage, which also affects biomolecules other than lipids, i.e., DNA and proteins. As will be discussed further on, the +2-oxidation state (ferrous) is the most dangerous free form of this metal and can participate as a catalyst in ROS formation reactions [18,25]. Nonetheless, given its metabolic importance, iron deficiencies are also associated with severe detrimental conditions, often resembling the widespread anemia disorders that, in turn, may increase the risk of developing complications, mainly affecting cardiovascular, nervous, and immune systems [26]. Overall, iron cellular homeostasis, as well as systemic metabolism and total body iron balance, must be sensibly regulated by a complex protein network (and their genes), which has evolved to allow both its complete biological utilization (uptake, recycling and storage) and the preservation of biological systems [12]. Biological players orchestrating the homeostatic control of iron metabolism have therefore come under the spotlight for their possible role in the regulation of ferroptosis, both in health and disease [27]. In this scenario, ferroptosis is now framed as a type of downstream non-apoptotic cell death pathway resulting from oxidative and metabolic stress, where high levels of iron and lipid peroxidation are the main triggers among a sequence of factors. The depletion of intracellular glutathione (GSH) and an impaired activity of glutathione peroxidases (GPXs) comprise the biochemical framework of ferroptosis and distinguish ferroptotic cells, in which iron accumulation allows for the Fenton reaction-dependent oxidation of lipids [28].

## 3. Iron-Dependent Oxidative Stress

The scientific literature is full of evidence describing the detrimental biological effects of iron overload, unveiling the irreversible damages that various types of biomolecules and biostructures can undergo under this condition [29]. Indeed, it is well established that iron accumulation in cells and tissues can disrupt redox homeostasis, propagate ROS generation, and predispose to oxidative stress [18,23]. Moreover, we now also recognize that iron can trigger and regulate the ferroptotic pathway of RCD.

The concept of “oxidative stress” was first conceived in 1985 to highlight the imbalance in cellular redox homeostasis resulting from a predominance of prooxidants over antioxidants. In this condition, higher quantities of ROS can be generated than those physiologically produced by the aerobic oxidative metabolism [30]. In a broad sense, the term ROS comprises a number of oxygen-containing reactive species, i.e., superoxide (O_2_•−), hydrogen peroxide (H_2_O_2_), hydroxyl radicals (OH•), singlet oxygen (1O_2_), peroxyl radicals (LOO•), alkoxyl radicals (LO•), lipid hydroperoxide (LOOH), peroxynitrite (ONOO−) and others, occasionally also referred to as reactive oxygen metabolites (ROMs), reactive oxygen intermediates (ROIs) and oxygen radicals. Actually, not all ROS contain unpaired electrons (e.g., hydrogen peroxide), which is why they cannot be defined generically as free radicals [31]. Moreover, the ROS group includes species defined as reactive nitrogenum species (RNS) since RNS are oxygen-containing species [32]. Throughout respiration, superoxide and hydrogen peroxide are continuously generated and enzymatically detoxified by cells, giving rise to a kind of cellular redox equilibrium under physiological conditions involved in redox signaling [33]. Superoxide dismutases (SODs), catalases (CATs), glutathione peroxidases (GPXs) and peroxiredoxins (PRXs) are ubiquitously expressed in aerobic organisms and are the main enzymes responsible for protection and detoxification from these ROS [34]. 

A key point is that the redox-active LIP is concerned with the formation of highly reactive free radicals. Indeed, the deregulation of cellular iron homeostasis resulting in increases in the LIP has definitively provided evidence of its involvement in the development of oxidative stress [15]. The chemical property that makes iron essential in biological systems—the ability to switch between Fe^2+^ and Fe^3+^ forms rapidly and easily—makes this metal paradoxically dangerous in aqueous biophases when unbound to proteins [18,29]. As a consequence, the LIP amount is normally subjected to strict intracellular regulations, i.e., it is either directly utilized in metabolic processes or sequestered in non-toxic forms [35]. It should also be remarked that iron overload is not always toxic per se, with proteins naturally predisposed to its sequestration, such as ferritin and hemosiderin, being capable of a safe iron storage capacity [36]. In addition, excess iron can also safely accumulate in lysosomes [37]. Intracellularly, ferritins allow for the formation of large biocompatible iron deposits characterized by a redox-inert state [38,39]. Due to their metabolic role in signaling mechanisms, superoxide and hydrogen peroxide are only moderately reactive compared to other much more dangerous ROS [40]. For instance, hydroxyl radicals are non-selective, very highly reactive species that can attack a variety of organic biomolecules [41]. Given their own highly oxidizing nature and short life span, hydroxyl radicals are generated in situ under specific conditions requiring, among other factors, the presence of free iron, thereby supporting the role of unshielded redox-active iron as a central regulator in ROS-induced cytotoxicity [15]. The link between iron and oxidative stress lies in so-called Fenton chemistry, in which free iron is recruited to generate additional highly reactive ROS. Indeed, the Fenton reaction strictly requires the presence of free iron in aqueous biophases in the ferrous oxidation state and is nowadays regarded as a biological prooxidant damage initiator [42]. The name comes from observations made by Fenton in the 1880s that a combination of acidified iron and H_2_O_2_ produces a strong oxidant [43]. Interestingly, transition metals other than iron endowed with biological functions (e.g., copper) and capable of exchanging electrons can correspondingly participate in Fenton-like reactions, propagating oxidative stress [44]. As expected, the Fenton reaction can have various implications in biology, being based on chemical species naturally present in cells under in vivo conditions. More broadly, the generation of ROS catalyzed by the presence of iron is part of a process known as Fenton and Haber–Weiss chemistry, where the Fenton reaction is defined as the reaction of ferrous iron and hydrogen peroxide with a contextual production of ferric iron and hydroxyl radicals. In turn, hydroxyl radicals can react with other peroxides to produce superoxide. Then, superoxide reacts again with peroxide, and both the hydroxyl radical and hydroxyl anion are formed. This final part of the process is recognized as the “Haber–Weiss reaction” [45]. Therefore, the Waber–Weiss reaction can further strengthen the oxidative injury triggered by the Fe^2+^/Fe^3+^ redox semicouple, causing DNA damage, lipid peroxidation and protein oxidation [15,44]. Indeed, excess ROS is believed to be one of the primary causes of DNA molecular damage, resulting in the accumulation of a variety of structural lesions. For example, Fenton-induced ·OH radicals can rapidly react with both DNA bases and deoxyribose to produce over 20 major oxidation products, as well as strand breaks and DNA–protein crosslinks [46]. Further oxidative lesions to DNA are discussed in the next section. Lipid peroxidation refers to ROS attacks on lipid carbon chains, preferentially on PUFAs containing unsaturated bonds. Typically, this ROS-mediated process occurs in three steps: initiation, transmission and termination. In the initial phase, promoters such as the ·OH radical remove hydrogen from lipids to produce lipid-free radicals with a carbon nucleus. Then, lipid-free radicals quickly combine with oxygen to form lipid peroxy radicals. At this point, the chain reaction goes on to produce new lipid-free radicals and lipid hydrogen peroxide through hydrogen removal from other lipid molecules [47]. As far as proteins are concerned, ROS can oxidize and damage amino acid side chains and protein backbones, especially interacting with sulfur-containing amino acids, i.e., cysteine and methionine. Under oxidative conditions, cysteines are converted into disulfides and then oxidized into sulfonic acid derivatives. Protein oxidation can also take place through the direct carbonylation of the side chains of lysine, arginine, proline, and threonine residues. Together with possible crosslinking, all these redox alterations can ultimately result in significant structural changes to the protein that may lead to loss of function [48].

## 4. Oxidative Stress, Iron and Cancer

### 4.1. ROS and Cancer 

An imbalance between the production of ROS and antioxidant defenses has been shown to be correlated with the development of many morbid conditions. Long-term oxidative stress is now accepted as an important promoter of human diseases, including neurodegenerative disorders, diabetes, cardiovascular illness, and, of course, cancer [49]. Healthy cells compared to cancer cells show a low-level steady state of ROS, associated with higher levels of reducing equivalents [50]. Accordingly, clinical studies have revealed that a low antioxidant status and increased oxidative stress levels are detected in cancer patients [51]. As a consequence, the detection of biomarkers arising from oxidative stress conditions is acquiring increasingly relevant diagnostic potential. In this context, products from lipid peroxidation (e.g., malondialdehyde) or DNA damage (e.g., 8-hidroxy-2-deoxyguanosine 8-OHdG), together with other biomarkers (e.g., antioxidative enzymes), can be investigated for a better understanding of the involvement of oxidative stress in cancer pathophysiology [49]. Prolonged exposure to oxidative damage and the persistent action of endogenous ROS can in fact cause an accumulation of damage to genetic material, with possible carcinogenesis. Although not all ROS affect DNA (superoxide and hydrogen peroxide at physiological levels do not react with DNA), it has been proven that hydroxyl radicals and other more reactive ROS can act directly on DNA, playing a detrimental role in damaging nucleic acids [52]. In these conditions, purine and pyrimidine bases can produce 20 different oxidative products, which are in turn implicated in subsequent events leading to mutations in genomic DNA, oncogene activation and cancer promotion [53]. In addition, ROS have been reported to cause a variety of lesions to DNA structure (such as base and/or sugar alterations/modifications, sugar–base cyclization, DNA–protein crosslinks, and intra- and interstrand crosslinks), which in turn can result in DNA strand breaks [54,55]. Therefore, oxidative stress may cause carcinogenesis not only by mutating nuclear or mitochondrial DNA but also by causing structural damage to intracellular lipids and proteins [56]. In breast cancer, oxidative stress may play a central role in tumor onset but can also be connected with age-dependent differences in gene expression and cancer biology [57]. In particular, oxidative stress pathways throughout cell immortalization and transformation have been related to both breast cancer heterogeneity and clinical prognosis [58]. Moreover, clinical evidence suggests an increase in lipid oxidation in patients with breast cancer compared to healthy controls [59]. Lipoperoxidation seems to also be increased in patients with prostate, bladder, liver, ovarian and colorectal cancers, suggesting that elevated oxidative stress and low antioxidant levels could contribute significantly to tumor pathogenesis and progression [49]. Similarly, increased levels of 8-OHdG detected in patients with different tumor types are indicative of significant ROS-dependent DNA lesions [49,60].

### 4.2. Iron and Cancer

In this scenario, excess iron can play a critical role by catalyzing the formation of ROS through Fenton and Haber–Weiss chemistry, giving rise to prooxidative conditions closely associated with carcinogenesis. Thus, for its ability to participate in redox processes but also for its biological relevance in cellular proliferation, iron is believed per se to be a carcinogen and to enhance the risk of cancer occurrence [61,62]. Moving in this direction, in-depth investigations on the behavior of iron and the iron regulatory pathway have been conducted, as well as on the possible nutritional impact of dietary iron on cancer onset [63]. Overall, epidemiological studies support the association between excess iron intake and increased cancer incidence and risk; in line with this, experimental investigations suggest iron’s implication in cancer initiation and progression, as well as in metastasis formation [64]. Hence, it is now a well-established opinion that iron excess and accumulation could increase the risk of cancer, acting both as a promoter and as a definite tumor growth factor [65]. Already since the middle of the last century, a series of experimental studies have disclosed that an overload of iron, both dietary and parenteral, is related to the development of tumors [62,66,67]. In more recent studies, serum iron overload from dietary intake was perceived as a risk factor for breast cancer, possibly due to increased oxidative stress [68]. According to the reference intake levels of nutrients and energy for the Italian population (LARN) managed by the Italian society of human nutrition (SINU), the recommended intake levels for iron vary according to age and gender. On average, for an adult male, they are around 10 mg/day, but up to 18 mg/day may be necessary for women of childbearing age and up to 27 mg/day for pregnant women. Intakes significantly higher than recommended are associated with a possible risk of cancer development [63].

Consistently, higher levels of dietary iron have also been associated with the development of colorectal cancer [69]. Moreover, patients with colorectal cancer, as well as cancer cells themselves, have deregulated iron metabolism, confirming this metal’s direct involvement in cancer development and progression [70,71]. More generally, current dietary habits associated with iron overload can be associated with an increased risk of developing neoplasms because of the induction of iron-mediated protumorigenic pathways, including hyperproliferation [64]. An intracellular increase in LIP levels through iron deregulation is among the foremost factors triggering sustained oxidative stress and potential neoplastic transformation [72]. Accordingly, altered iron homeostasis resulting from enhanced metal demand is a common feature of several malignant cells, which reprogram their metabolism in a multifaceted process to meet needs related to tumor phenotypes [73]. In fact, increased expression of the transferrin receptor (TfR) is now well established in a variety of cancers, as is a concomitant downregulation of the iron exporter ferroportin (FPN) [74,75]. It is noteworthy that cancer cells self-produce hepcidin, whose main biological effect is to increase the amount of LIP through the degradation of ferroportin [76]. In this way, elevated iron meets the metabolic requirements for malignancy for unlimited proliferation but also for redox signaling to withstand growth and spread [73,77]. This also explains why the use of effective iron chelators can be therapeutic in the treatment of some forms of cancer—a curative option that deserves further exploration for the development of new therapeutic strategies [78,79,80,81]. Furthermore, TfR and FPN could represent potential therapeutic targets and important biomarkers of some tumor pathologies. Their central role in the regulation of iron homeostasis also associates these proteins with the regulation of ferroptosis and the development of specific tumor microenvironments (TMEs), which in turn can have significant effects on tumor progression and responsiveness to therapies [76,82]. 

Overall, a growing body of evidence now suggests that tumor cells can benefit from deregulations in iron homeostasis aimed at increasing iron availability. An imbalance of cellular iron homeostasis is exploited for both the onset and progression phases of cancer. Remaining within this study’s scope, one of the most interesting findings has been the observation that the iron-responsive element-binding proteins (referred to as IRE-BP)—simply known as Iron Regulatory Proteins (IRPs)—could have regulatory functions contributing to tumor progression independent of their primary roles as master regulators of iron homeostasis [78]. For instance, though the expression and activity of IRPs in cancer differ by tumor type, the overexpression of IRP2 has been observed at a preclinical level in many cancer cells and is correlated with altered iron phenotypes, while the downregulation of IRP1 has been linked in clinical studies with a poor prognosis of cancer progression [83,84]. Moreover, IRP1 was found to be critical in regulating iron homeostasis and ferroptosis in melanoma cells [85]. Additionally, under conditions of increased ROS production, iron–sulfur clusters assembled in the IRPs can be destabilized, contributing to protein degradation, suggesting of a further possible dynamic interaction between IRP activity, iron homeostasis, and ROS levels [86]. 

## 5. Hallmarks of Ferroptosis from a Cytomorphological Perspective 

It has been assumed that ferroptosis and the other types of regulated cell death (RCD) are very different from cellular and biochemical perspectives [87]. Indeed, ferroptotic cells exhibit unique and special hallmarks [88]. No nuclear morphological changes, DNA fragmentation and/or chromatin condensation, as well as any caspase activation, have been observed. Light and electron microscopy observations also confirm that there is something very different between the known PCD pathways [89,90]. From a cytomorphological standpoint, ferroptosis is closer to necrosis than other RCDs, especially due to its cytoplasmic membrane appearance throughout the process, while it differs completely from apoptosis [91]. In fact, plasma membrane swelling and cell enlargement, with subsequent membrane rupture, are normally detected in ferroptotic cells, as typically occurs throughout cellular necrosis [92]. Both are usually accompanied by the onset of inflammation, which is not observed during apoptosis. In compliance with this, immune cell infiltration has been detected in tissues affected by ferroptotic damage [93]. This is the reason why ferroptosis observed in vitro and in tissue can hardly be distinguished from necrotic morphology. Within the cell, an important alteration in the mitochondrial ultrastructure can be observed through electron microscopy during ferroptosis, such as a reduction in the mitochondrial volume, an increase in the mitochondrial membrane density, the disappearance of mitochondrial cristae and a rupture of the outer membrane caused by the lethal accumulation of lipid hydroperoxides [1]. These alterations are connected to mitochondrial dysfunction and significant changes in biochemical markers [94]. With regard to the appearance of nuclei, although ferroptosis is notoriously accompanied by oxidative stress and DNA damage, no significant nuclear or chromatin alterations have been reported to date. However, some authors described a lack of chromatin concentration, which instead is a morphological indicator of apoptosis [4]. To recap, distinctive morphological features in ferroptotic cells concern mainly the condensation of mitochondria and a loss of integrity of the outer membrane. 

Notably, very recently, significant progress has been made in ferroptosis research based on the design of specific fluorescent probes and their bioimaging applications. Indeed, fluorescence imaging can provide a major contribution towards understanding the changes in crucial (bio)molecules and cellular microenvironments throughout ferroptosis. Moving in this direction, useful “visual” evidence concerning key factors of ferroptosis, including, among others, ROS and iron levels, is becoming available [95,96]. Hence, given the biochemical complexity of the ferroptosis process, the further development of fluorescence imaging techniques could be decisive for the optimization of protocols of detection and characterization. 

Beyond cytomorphological analysis, established methods for ferroptosis detection are based on the determination of specific hallmarks, i.e., cellular iron accumulation, glutathione (GSH) depletion, lipid peroxidation and ROS generation [91]. In addition, alternative methods like the one for evaluating the native enzymatic activity of glutathione peroxidase 4 (GPX4) are being developed and validated [97]. As discussed extensively in the next section, due to its redox ability and GSH recruitment, GPX4 is the main ferroptosis inhibitor and can prevent its activation. Thus, the evaluation of its biological activity can provide useful information for characterizing ferroptosis. Correspondingly, the expression, activity and cellular localization of ferroptosis suppressor protein 1 (FSP1) could provide further information on ferroptotic flow and its activation/deactivation. Indeed, FSP1 is a glutathione-independent enzyme which acts in parallel with GPX4 as another key ferroptosis inhibitor [97].

## 6. Molecular Mechanisms of Ferroptosis

### 6.1. Ferroptosis Inducers (FINs) 

Unlike so-called accidental cell death (ACD) or unprogrammed cell death—simply known as necrosis—which occurs in an unplanned manner, usually in response to acute cellular injuries, RCD requires the execution of ordered molecular pathways according to well-defined sequences [90]. Also, structured signaling cascades in response to definite types of stress must occur until the activation of specific molecular effectors (e.g., effector caspases throughout apoptosis) [98]. Compelling experimental evidence now proves that ferroptosis is a unique type of RCD very different from apoptosis, autophagy and others [21,87,88,99]. In fact, it has been demonstrated that ferroptosis can be hindered by iron chelators and lipophilic antioxidants, which selectively interfere with the pathways triggering iron-dependent cell death but not with the activation and execution of the other programmed cell deaths [9]. Thus, selective and effective inhibitors of apoptosis, autophagy and necrosis (i.e., caspase inhibitors and necrostatins) do not produce significant effects on ferroptosis [87].

In line with this, as already discussed, the treatment of tumorigenic cells with erastin selectively triggers ferroptosis without evidence of other hallmarks of RCD [1,3] (the molecular structure of erastin is shown in Figure 3a). Erastin activates voltage-dependent anion channels (VDACs) and inhibits system XC− (cystine/glutamate antiporter), thereby blocking cellular cystine intake and glutamic acid efflux in an ATP-dependent manner [1,100]. In more detail, it has been conjectured that erastin interacts with one of two proteins—a light chain (SLC7A11/xCT) and a heavy chain (SLC3A2)—linked by a disulfide bridge and forms the functional transmembrane heterodimer of this antiporter system [1,101]. In cells, a decreased availability of cystine (the oxidized form of cysteine) impairs the activity of both GSH synthase and glutamate cysteine synthase, physiologically producing GSH from glutamate, glycine and cysteine. The resulting depletion of GSH, the most abundant non-protein thiol engaged in defense against oxidative stress, significantly reduces the antioxidant capacity of cells and paves the way for ROS production, lipoperoxidation and diffuse oxidative stress [102]. Oxidized GSH is in fact essential for the activity of glutathione peroxidase 4, also known as GPX4, which is encoded by the GPX4 gene [103]. GPX4 belongs to the family of glutathione peroxidases (GPXs), which consists of eight known mammalian isoenzymes (GPX1–8) [104]. Specifically, GPX4 is a selenium-containing phospholipid hydroperoxidase committed to cell protection against membrane lipid peroxidation [105]. Besides GSH, other small thiol metabolites such as cysteine and homocysteine are also able to serve as cofactors for GPX4, nonetheless with a lower affinity for the enzyme than GSH [106]. In terms of its biological role, since the discovery of ferroptosis, GPX4 activation has been regarded as an antagonist mechanism of this type of RCD. Conversely, the inactivation of GPX4 (e.g., as in the case of GSH depletion) leads to lipid peroxide bioaccumulation by reducing cellular antioxidant resistance, thereby prompting ferroptotic cell death [107,108].

Consistently, other treatments that reduce cellular GSH levels can trigger the ferroptotic pathway by roughly the same mechanism, such as treatment with L-buthionine sulfoximine (BSO) or with sorafenib and artesunate [109,110,111]. On the other hand, enhanced levels of cystine/cysteine, as well as enhanced cysteine synthesis, can rescue cells from erastin-induced ferroptosis [112]. As well as the fact that no apoptotic hallmarks can be observed following erastin-induced cell death, apoptotic inhibitors cannot prevent erastin-dependent ferroptosis [87]. As far as other ferroptotic inducers are concerned, RAS-selective lethal 3 (RSL3, see Figure 3a), discovered together with erastin at the beginning of the 2000s, behaves as an electrophile that directly interacts and inactivates GPX4 through nucleophilic addition to selenocysteine at the active site [47]. More recently, altretamine (also called hexamethylmelamine), approved by the FDA as an antineoplastic agent in 1990, has been recognized as a novel ferroptotic inducer endowed with the ability to inhibit GPX4 [113]. Still, at Columbia University, starting from a cytotoxic agent known as CIL56, Prof. Stockwell’s group developed in 2016 a compound named FIN56 (ferroptosis inducer 56) that retained oncogenic RAS selectivity through ferroptosis activation. In particular, FIN56 exhibits ferroptosis-inducing activity via two different pathways involving both GPX4 degradation and CoQ10 depletion. Although the mechanisms behind GPX4 degradation are still unclear, FIN56-mediated CoQ10 depletion seems to be caused by interference with the mevalonate pathway and isoprene synthesis, finally leading to the formation of the carbon skeletons of its precursors [112]. Furthermore, FIN_O2_, a stable endoperoxide containing 1,2-dioxolane, is believed to be a potent ferroptosis inducer. However, its activity does not seem to occur through interference with amino acid metabolism (as erastin does) or through the direct inhibition of GPX4 (as RSL3 does). Researchers have conjectured that FIN_O2_ is able to indirectly hinder the activity of GPX4, while it can oxidize ferrous iron directly to produce ROS, as in Fenton chemistry. In this way, lipids would also be widely oxidized, with a subsequent induction of ferroptosis [114,115]. 

Obviously, among the inducers of ferroptosis, the cellular free iron pool (LIP) should rightly be considered, being able to trigger and propagate prooxidative reactions, which in turn can favor the intracellular accumulation of lethal lipoperoxides [116,117]. In short, recognized ferroptosis inducers (FINs) used in the experimental field can act via different molecular mechanisms while, in the end, producing similar biological effects: the accumulation of lipid-reactive oxygen species, the propagation of oxidative stress and damage to cell membranes. Indeed, it is broadly accepted that the oxidation of the carbon chains of membrane complex lipids enables ferroptosis in cells [47,118]. Therefore, FINs can be framed as a promising solution to induce cell death and overcome drug resistance in certain diseases. As will be explored further later, drug-like agents capable of selective proferroptotic effects in cancer cells are regarded by scholars as potential key contributors in the development of a new anticancer strategy [119,120]. Aiming to target the ferroptosis pathway and possibly strengthen existing therapies, research is very propositive and could provide new therapeutic options for the treatment of cancer. Based on our current knowledge and what we have discussed, FINs can finally be classified into four categories, considering their mode of action: SLC7A11 inhibitors (I), GPX4 blockers (II), inducers of GPX4 protein depletion (III), iron oxidizers and indirect GPX4 inactivators (IV) [121]. 

### 6.2. Inhibitors of Ferroptosis

The study of possible inhibitors of ferroptosis offers a complementary view to focusing on enzymes and pathways that are implicated in its regulation [122]. As expected, agents that act by enhancing the cell’s antioxidant defenses can have an antiferroptotic effect [123]. Thus, strengthening the functioning mechanism of the GPX4 system (through the availability of cysteine, the synthesis of GSH, the formation of selenocysteine, etc.) can prevent or at least limit ferroptosis [106]. Likewise, free radical scavengers and antioxidants, including vitamins that can protect lipids from oxidation, may play a role in this regard. Radical-trapping antioxidants (RTAs), as well as free radical scavengers, can stop free radical chain reactions that cause the accumulation of lethal molecules. Among the best known, tocopherols, including α-tocopherol (vitamin E; see Figure 3b), have been demonstrated to protect membranes from lipid peroxidation and to act as ferroptosis suppressors in vitro and in vivo [109,118]. And, as expected, vitamin E and GPX4 synergistically inhibit lipid oxidation, both in cells and animal models [124]. Ferrostatin (Fer-1, see Figure 3b), uncovered through the high-throughput screening of small-molecule libraries, is certainly among the best-known synthetic antioxidants and is very effective and selective in inhibiting ferroptosis [125]. It is able to suppress erastin-induced ferroptosis by acting via a reductive mechanism to prevent damage to membrane lipids and thereby inhibit cell death. In particular, as an established RTA, Fer-1 scavenges hydroperoxyl radicals, limiting lipid peroxidation. Nonetheless, Fer-1 has also been demonstrated to reduce the cellular LIP [126]. Liproxstatin (Lip-1) was identified in a similar way and behaves as an antioxidant cytoprotector [127,128]. Both Fer-1 and Lip-1 are much more effective antioxidants than vitamin E in experimental models, and their mechanism of action has underscored again the role of lipid autooxidation during ferroptosis [129]. 

Interestingly, iron(II) and iron(III) chelators deserve particular consideration when delving into ferroptosis inhibition. Indeed, we have highlighted iron’s role as the main contributor to the induction and dissemination of oxidative stress at a cellular level [15,27]. To date, several iron chelators are under investigation, while others have already achieved clinical employment for the treatment of iron overload disorders (see Figure 3b for molecular structures of Deferoxamine and Deferiprone, among the best known iron chelators) [24,80]. From an anticancer perspective, the paradox lies in the fact that if a decrease in the LIP through iron chelation hinders ferroptosis as a mechanism of RCD, decreased iron availability could result in direct antiproliferative effects [119]. In this context, the potential applications of synthetic compounds, which, rather than chelating iron, interfere directly with the production of ROS via Fenton chemistry, are also very interesting. For instance, nitroxides can reduce ROS-dependent oxidative stress, in particular inhibiting the formation of hydroxyl radicals even in animal models through the oxidation of redox-active iron [130].

More recently, ferroptosis suppressor protein 1 (FSP1), already known as apoptosis-inducing factor mitochondria-associated 2 (AIFM2), was recognized as an effective ferroptosis inhibitor. Its enzymatic activity is considered to parallel that of GPX4 in the control of ferroptosis [131]. FSP1 is engaged in the plasma membrane as an oxidoreductase, using coenzyme Q10 (CoQ10) and NADPH to function as a lipophilic radical scavenger [132]. Furthermore, assuming the involvement of lipoxygenases (LOXs) in the enzymatic process of lipid oxidation, LOX inhibitors should also be included among the agents capable of mitigating ferroptosis [133]. In Table 1, the main features of ferroptosis inducers (FINs) and inhibitors are summarized. 

## 7. Iron-Dependent Lipid ROS Generation and Leading Molecular Mechanisms of Ferroptosis

As anticipated, beyond the impairment of the detoxification mechanisms causing an accumulation of lipid ROS, lipid oxidation leading to ferroptosis can depend on different enzymatic and non-enzymatic molecular mechanisms, all requiring the presence of iron [21,91,99]. Specifically, enzymatic processes cause iron-containing lipoxygenases (LOXs), while non-enzymatic processes lead to iron-dependent Fenton reactions (see Figure 4 for a schematic view of the entire process). 

Overall, growing evidence is in favor of a general alteration in both lipid metabolism and in the structure of the amphiphilic lipids of plasma membranes as drivers of ferroptosis [134,135]. LOXs are a family of (non-heme) iron-containing enzymes involved in the metabolism of polyunsaturated fatty acids (PUFAs) [136]. LOXs catalyze lipid dioxygenation into cell signaling molecules involved in the metabolism of eicosanoids that serve diverse roles as autocrine, paracrine and endocrine signals. Each LOX contains one iron ion as a cofactor, which is neither coordinated by porphyrins nor by iron–sulfur clusters. They work as catalyst coordination complexes in redox processes, adopting an iron cation packaged in the active site, molecular oxygen and a lipid substrate undergoing peroxygenation to produce organic peroxide and hydroperoxides [137]. During oxidative stress, PUFAs are the preferential targets of lipid peroxidation in the major structural lipids of membranes [138]. In turn, among PUFAs, arachidonate and its analogues resulting from the same anabolic pathways are particularly vulnerable to oxidation, so much so that isoforms of arachidonate lipoxygenase (ALOX), including ALOX5, ALOX12, ALOX15, ALOX15B, and ALOXE3, are considered key enzymes in controlling ferroptosis [91]. The crucial functions of LOXs in ferroptosis are supported by evidence that, though several enzymes can oxidize fatty acids in cells, selective inhibitors of LOXs can prevent ferroptosis [118]. 

Moreover, an LOX-independent alternative mechanism involved in lipid peroxide generation based on cytochrome P450 oxidoreductase (POR) activity has recently been recognized [139]. In this framework, members of the long-chain Acyl-CoA synthetase family (ACSLs) are also regarded as potential proferroptotic regulators by recruiting long-chain fatty acids to enrich cellular membranes with PUFAs [118,140]. Mainly localized in the endoplasmic reticulum and mitochondrial outer membrane, ACSLs catalyze the formation of acyl-CoAs from fatty acids to start new anabolic processes, thereby shaping the lipid composition of plasma membranes [140]. However, among ACSL isoforms, only ACSL4 has a direct correlation with ferroptosis that has been proven so far [141,142]. 

It is noteworthy that, despite the fact that ferroptosis can be considered an LOX-dependent process, not all inhibitors of LOXs can save cells from ferroptosis [133]. Accordingly, as already discussed, many agents inhibiting ferroptosis are fundamentally radical-trapping antioxidants (RTAs), i.e., lipophilic radical traps such as α-tocopherol, ferrostatin-1 and liproxstatin-1 [129]. RTAs can protect cells against iron-dependent free radical chain reactions and the detrimental effects of oxidative stress propagation, but they do not interfere with the LOX-dependent formation of lipoperoxides [1,109]. In support of these findings, it has also been demonstrated that antioxidants acting predominantly on lipid substrates can counteract in different ways the propagation of oxidative stress, even in the absence of an active GPX4/GSH system [132,143]. It is as if to say that ferroptosis is predominantly driven by lipid autoxidation rather than LOX-controlled lipid peroxidation. More precisely, though LOXs are engaged as early inducers of the cellular pool of lipid peroxides, subsequent lipid autoxidation seems to be the effective driving force of the whole ferroptotic process [133]. In this situation, the pool of cellular labile iron (LIP) and Fenton chemistry are assumed to be the main determinants (see Section 3) [18]. This is the reason that, both bound to proteins and free in cellular biophases, iron is considered the master regulator at the core of the molecular mechanism compelling ferroptotic cell death [6]. As a consequence, hallmarks of cellular iron homeostasis (i.e., iron uptake, storage, utilization and export) can have a significant regulatory impact on ferroptosis [27,144]. Eventually, the entire sequence of events resulting in ferroptosis can be exacerbated by the breakdown of the GSH/GPX4 antioxidant system, as underlined by the mechanisms of action of the two conventional ferroptosis activators, i.e., erastin and RSL3 [103]. 

What exactly happens and how it correlates lipid oxidative stress with ferroptotic cell death is still indefinite. It is believed that oxidative alterations in lipid bilayers most likely cause a destabilization of the membrane, resulting in irreversible damage such as the formation of pores and micelles. This could in turn amplify damage in a spiral of oxidative events from which the cell would have no escape [121]. In addition, the decomposition of oxidized lipids into cytotoxic aldehydes (e.g., malondialdehyde) can cause the inactivation of proteins and contribute to cell death. It is noteworthy that the understanding of these molecular mechanisms leading to ferroptosis is thoroughly correlated with the activity of the repair mechanisms of biological membranes and could provide new insights into controlling this type of RCD [145]. Indeed, enzymatic machineries engaged in plasma membrane repair can rescue cells from ferroptosis under stress conditions, as demonstrated by the role of the endosomal sorting complex required for transport (ESCRT), whose overexpression is correlated with membrane recovery and the repair of pores [146,147]. 

## 8. Ferroptosis and Cancer: Prospective Druggable Targets for Novel Therapeutic Interventions

### 8.1. Ferroptosis, a Molecular Mechanism of Cancer Cell Death or Survival?

Looking at the possible development of effective antitumor therapies, the discovery of a new mechanism of RCD has represented an attractive opportunity to be explored in order to fight malignancies. This is of special concern, especially for the treatment of aggressive and chemoresistant cancer phenotypes with limited therapeutic options [148]. Therefore, understanding the molecular mechanisms behind ferroptosis could provide key insights to develop unconventional strategies for cancer therapy. As discussed so far, the current findings have highlighted significant alterations in lipid metabolism during ferroptosis and, among others, have led to promising insights into the interplay between iron homeostasis and metabolism in cancer cells [119,120]. Their interaction can remarkably influence cancer onset and development and metastasis formation, as well as therapy-related aspects, including cancer resistance and immunity [149]. However, despite the outstanding scientific advances highlighting connections concerning ferroptosis and human diseases, to date, the interplay between ferroptosis and cancer still remains unclear and very intricate [99]. Several scientific studies have described ferroptosis as a cellular protective mechanism for evading the development of cancer, while it is equally recognized that ferroptosis may be a defense device for cancer cells to counteract oxidative stress and metabolic alterations [139,150,151,152]. This midpoint situation is very reminiscent of autophagy in cancer, believed to be a mechanism of cell survival in many tumor phenotypes, but when dysfunctional, it is considered a possible RCD mechanism to be considered among the therapeutic options [153,154]. 

Moreover, it should be noted that the specific role of ferroptosis and its implications as a mechanism of cell death or cell survival could be closely related to the specific type of tumor and to the ever-changing typology of the TME that is generated. The exploration of immune cell responses in the TME to the metabolic features of tumor phenotypes and the chemical properties of the microenvironment itself (e.g., changes in the redox conditions, the presence of toxic oxidized species, oxygen and nutrient deprivation) has just started. From this perspective, immune cells in the TME could exhibit individual sensitivities to ferroptosis, and this in turn could have serious implications for therapeutic responses. For instance, ferroptosis activation could be a weakness of activated cytotoxic T lymphocytes, while ferroptosis inhibition could help their survival and antitumor effects [155]. Thus, modulators of ferroptosis, such as inducers (FINs) or inhibitors, might have profound impacts on cancer immunity that need to be precisely evaluated. For this reason, the opportunity of selectively targeting ferroptosis in immunocompetent cells of the TME is nowadays of great interest from a therapeutic point of view [156]. 

### 8.2. Ferroptosis Regulation by Oncogenes and Oncosuppressors

Ferroptotic machinery can be finely regulated by both oncosuppressors and oncogenes, which act as suppressors or promoters of ferroptosis, respectively [91,157]. The dual role of ferroptosis in cancer is widely reported in the literature. Among oncosuppressors, noteworthy is *p53*, which represents the most frequently mutated gene in numerous tumors. The main mechanism through which *p53* stimulates ferroptosis, assumed to be a mechanism of cell death, is the inhibition of SLC7A11 carrier expression, leading first to cysteine depletion and then to glutathione deficiency. From this perspective, the *SLC7A11* gene exhibits increased expression in various human cancer types, causing the inhibition of ferroptosis [158]. Thus, the oncosuppressor *p53* can reduce the expression of the SLC7A11 antiport, thereby inducing ferroptosis via LOX-independent pathways [119,159]. Moreover, it has been reported that *p53* can induce the production of lipid hydroperoxides to enhance ferroptosis by increasing the expression levels of LOXs [160]. However, other studies have reported that *p53* plays an antiferroptotic role, highlighting its possible dual role in this process [161,162]. An additional tumor suppressor involved in ferroptosis regulation is *BRCA1-associated protein 1* (*BAP1*), which is mutated in many cancer types. This gene encodes a nuclear deubiquitinating enzyme responsible for reducing the ubiquitination of histone H2A (H2Aub) on chromatin [163]. *SLC7A11* has been identified as a target gene of *BAP1*. In fact, several studies demonstrate that *BAP1* reduces the expression of SLC7A11, with a consequent increase in lipid peroxidation and therefore ferroptosis induction [164,165,166]. The other way around, many oncogenes can stimulate cancer progression through ferroptosis evasion [21,167]. Indeed, oncogenic *KRAS*, closely related to lung adenocarcinoma development, induces an upregulation of the carrier SLC7A11 and, therefore, an increase in the intracellular concentration of cystine and cytoprotective glutathione in conditions of oxidative stress [168,169]. In KRAS-mutant lung adenocarcinoma, the inhibition of *SLC7A11* reduces tumorigenesis, demonstrating its role in ferroptosis evasion under oxidative stress [170]. Among other oncogenes contributing to tumor initiation and progression, *PI3K* has been revealed to induce the activation of mTOR complex 1 (mTORC1), which is responsible for the inhibition of the ferroptotic pathway through different mechanisms, including an increase in GPX4 protein synthesis [119,149]. 

### 8.3. Ferroptosis as a Therapy Target for Upcoming Anticancer Strategies

With regards to the possible activation of proferroptotic effects and the identification of novel druggable targets from an antitumor perspective, growing evidence suggests that ferroptosis is a critical tumor suppression mechanism [119]. To date, a large amount of data are available for FINs belonging to classes I and II, which have also been tested in vivo [91]. Although erastin is one of the most used FINs in vitro, in vivo, its applications are limited by its reduced metabolic stability [121]. However, imidazole ketone erastin (IKE)—a more potent and stable derivate—has been demonstrated to significantly reduce tumor growth in vivo [171]. Sulfasalazine, which is regarded as a class I FIN, is an anti-inflammatory drug approved by the FDA to treat Crohn’s disease, ulcerative colitis and rheumatoid arthritis and has been shown to reduce tumor growth in vivo [172]. Interestingly, it is currently being tested in a clinical trial for recurrent glioblastoma in combination with radiotherapy (this and some other clinical trials encompassing ferroptosis are reported in Table 2). Moreover, the kinase inhibitor sorafenib was also shown to induce ferroptosis by acting against the SLC7A11 carrier [173]. RSL3, ML210 (and its derivate JKE-1674) and ML162 are the most studied compounds belonging to class II FINs, acting as GPX4 covalent inhibitors. They have mainly been tested so far in cellular preclinical studies due to their instability in physiological conditions [174]. Indeed, overcoming their pharmacokinetics and selectivity issues remains a challenge in the development of new GPX4 inhibitors for cancer therapy. However, some already known anticancer agents, i.e., altretamine and withaferin A, have been recently recognized as GPX4 activity inhibitors, offering an alternative for the targeting of GPX4 in vivo [113,152]. Overall, a great deal of effort to design and develop anticancer drugs based on ferroptosis induction is currently underway. As the discovery of FINs (starting with erastin and RSL3) is correlated with an opportunity to inhibit oncogenic RAS-mutant cells, and considering that a recent in-depth analysis showed that Ras mutations and incidence exhibit differential coupling to specific cancer phenotypes, at least in principle, definite cancer cells should be more susceptible to the action of erastin and congeners [169]. Nonetheless, meta-analysis and cancer genetics databases show a wide distribution of tumor types with Ras mutations. In line with this, approximately a third of all human cancers, including a high percentage of pancreatic, lung, and colorectal cancers, are driven by mutations in RAS genes [175]. This has been validated by several experimental models based on the use of tumor phenotypes of different histological origins, which have been proven to be sensitive to the action of FINs as potential candidate anticancer drugs [176]. However, cancer cells are generally more susceptible to ferroptosis than normal cells for inherent reasons, including their high ROS production, elevated intracellular iron level, and dependence upon oncogenic signal transduction crosslinked to the ferroptotic pathway [28,40,49,73].

One of the main limitations of anticancer therapies is the onset of resistance phenomena, which make clinical interventions ineffective [79,177]. With due caution, researchers can now exploit an additional weapon that consists of (re)activating ferroptosis as a tumor suppression mechanism to overcome resistance. From this perspective, several FINs holding great potential for forthcoming cancer therapy are being explored for their ability as selective activators of ferroptosis in specific cancer phenotypes [176]. 

In parallel, light is being shed on the molecular mechanisms that allow cancer cells to escape ferroptosis. Hopefully, all this information will allow for the identification of increasingly suitable targets to design and develop precise drug-like compounds endowed with effective anticancer activity [119,178]. Moving in this direction, novel anticancer chemotherapeutics in progress act with a multimodal action, triggering multiple cell death pathways, including ferroptosis. This approach could limit the onset of resistance [179]. Several anticancer metal-based platforms designed for prospective biomedical applications behave as multitarget agents that work in this way, giving rise to multiple intracellular interactions by virtue of their molecular features and adaptability [180]. Among these, many ruthenium- and other metal-based complexes developed in recent decades as alternatives to platinum-based compounds are emerging as potential anticancer drugs and have, in some cases, reached clinical trials [181]. Interestingly, ruthenium shares many physico-chemical similarities with iron from a biological point of view [49,182]. Furthermore, some ruthenium-based drug candidates are capable of triggering oxidative stress at a cellular level, and this may have a considerable impact on their antiproliferative mechanism [183]. In principle, it could be feasible to explore the ruthenium/iron/ROS/ferroptosis axis in order to outline new options for the treatment of cancer, thus opening up new scenarios for the design of unconventional therapeutic strategies.

A further attractive strategy to boost ferroptosis induction could be a combination of conventional cancer therapies and FINs. This advanced approach could be useful in resensitizing resistant types of cancer to ferroptosis. For example, in response to radiotherapy, cancer cells adapt by enhancing expressions of SLC7A11 or GPX4 to fight ferroptosis induced by radiotherapy. This phenomenon could be overcome by the association between radiotherapy and FINs [184]. Similarly, FINs targeting SLC7A11 or GPX4 could sensitize cells to conventional therapies, including chemotherapeutics such as cisplatin, doxorubicin and gemcitabine [185,186]. This combined therapeutic approach has shown limited toxic effects on normal cells, demonstrating good tolerability in preclinical models [119,176,187]. 

### 8.4. Targeting Iron to Regulate Ferroptosis

Considering the central role of iron in the regulation of ferroptosis, tumor oncosuppressors and oncogenes interfering with iron homeostasis can play pivotal functions in cell fate decisions towards the stimulation of ferroptosis (cell death) or the inhibition of ferroptosis (cell survival). Indeed, as a unique form of programmed death featured by the cellular accumulation of iron-dependent lethal lipid peroxides, the molecular mechanisms governing ferroptosis inevitably intersect those regulating iron homeostasis [27,151]. Cancer phenotypes are naturally programmed to elude ferroptosis, as well as other types of RCD such as apoptosis, through several mechanisms ensuring cell survival and proliferation [188]. Understanding the latter, together with emerging knowledge on the regulation of ferroptosis itself, could offer researchers a very motivating background for the development of new therapeutic strategies. In some cases, it has been demonstrated that tumor cells can evade ferroptosis by restricting the LIP [119]. In line with this, proteins involved in the biosynthesis of iron–sulfur clusters, such as frataxin and cysteine desulfurase NFS1, are upregulated in different types of cancer cells, inducing a reduction in intracellular free iron levels and protecting cells from ferroptosis [189,190]. Conversely, it is also known that cancer cells share an altered metabolism characterized by a high amount of ROS, which could make cells more susceptible to ferroptosis [150,191]. Indeed, neoplastic types with limited therapeutic options such as small-cell lung cancer (SCLC) and triple-negative breast cancer (TNBC) exhibit susceptibility to ferroptosis due to their inherent distinctive metabolic characteristics, including increased intracellular levels of PUFA and LIP coupled with a reduced activity of the GPX4–GSH defense system [140,192]. Interestingly, these LIP adaptations occur as cancer cells attempt to reach the right iron homeostatic balance without compromising its availability, which is fundamental for their metabolism requirements [79]. However, molecular mechanisms that induce an increase in LIP can be considered proferroptotic mechanisms. In principle, drugs that have the potential to increase the amount of available free iron in both ferrous and ferric forms can promote iron-dependent oxidative stress conditions and direct the cell towards ferroptotic death [193]. Therefore, a new frontier of biomedical research in the oncology field is aimed at the possibility of selectively triggering or reactivating pathways leading to ferroptosis. In this direction, selective iron donors in tumor cells could be developed as agents capable of (re)activating ferroptosis. Paradoxically, this approach is conceptually in contrast with the traditional conception based on the inhibition of cancer cell growth and proliferation through the use of iron chelators, whose, together with ferroportin’s, overexpression can limit the invasiveness and malignancy of metastatic cells [80,108,194]. 

### 8.5. Lipid Druggable Targets to Control Ferroptosis 

Similarly, lipid synthesis, storage and degradation are closely associated with ferroptosis. Molecular mechanisms and enzymes that regulate lipid metabolism can become druggable targets to control the formation and accumulation of oxidized species that can eventually direct cancer cells towards ferroptosis [195,196]. Thus, exogenous agents able to modulate lipid metabolism can have profound effects on the regulation of ferroptosis [134]. Genetic screening and microarray analysis have recently endorsed the intimate connection between lipid metabolism and cell sensitivity to ferroptosis, strengthening the importance of pathways and enzymes involved in lipid biosynthesis, including ACLSs and LOXs [140]. They can influence the type and abundance of lipids in the plasma membrane, which in turn impact the type of lethal lipoperoxides and oxidized species that are formed and accumulated. In particular, an abundance of PUFAs in the plasma membrane that are particularly susceptible to oxidation is a key factor in determining cell sensitivity to ferroptosis [197]. In line with this, the application to cells of exogenous monounsaturated fatty acids (MUFAs), e.g., oleic acid (OA), which can displace PUFAs in phospholipids following activation by ACSL3, leads to the inhibition of ferroptosis [198]. Interestingly, acyl-CoA synthetase (ACSL) family members, including ACSL3, required for exogenous fatty acid activation, are deregulated in cancer, highlighting the significance of lipid metabolism in these processes, as well as the possibility of designing targeted therapeutic interventions [199]. In line with this, it has been demonstrated that the inhibition of specific desaturases engaged in MUFA synthesis can navigate cells towards ferroptosis by modulating the inclusion of unsaturated fatty acid chains in membrane phospholipids [200]. 

Moreover, we have previously shown the central central role of GPX4 in the detoxification of oxidized lipids by converting GSH into GSSG. Cancer cells tend to control redox balance to prevent excessive peroxidation, eventually facilitating tumor growth. Considering first the relevance of cysteine cellular intake and then of transsulfuration pathways in this process, their inhibition could represent another efficient strategy to stimulate ferroptosis in cancer cells, retracing the path of canonical ferroptosis inducers (i.e., erastin and RSL3) in blocking the XC−/GSH/GPX4 axis [201]. Thus, the antiporter system Xc- is a promising target in cancer cells for ferroptosis induction [202]. The challenge is represented by looking for selectivity of action in cancer cells. Remarkably, the regulation of lipid metabolism in connection with ferroptosis can also have major implications for tumor invasion and metastasis. Generally, metastatic phenotypes overexpress GPX4 and inhibit the activities of COXs and LOXs to limit lipid peroxidation levels. Recent findings connect these and other features to the metastatic ability of cancer cells [134]. It has also been reported that the overexpression of GPX4 decreases the metastatic colonizing capacity of some tumor phenotypes in relation to their ability to produce eicosanoids [203]. 

Hence, many aspects remain to be clarified. It is certain that the regulation of lipid metabolism represents another multifaceted factor which affects cancer progression. Its elucidation from a ferroptosis perspective could pave the way in the future for unconventional strategies for treating cancer. 

## 9. Concluding Remarks and Future Perspectives

Nowadays, we have ultimately assumed that ferroptosis is a definite type of RCD and that its occurrence is iron-dependent. Hence, it is usually accompanied by a large amount of iron accumulation and lethal lipid peroxidation in cellular plasma membranes. Meanwhile, biomedical studies over the last decade have provided compelling evidence supporting ferroptosis as a key pathogenic player in many diseases’ progression, such as various cancer types and neurodegenerative processes, as well as inflammatory and infectious diseases. Although some molecular pathways and process networks that characterize ferroptosis remain to be elucidated, considerable progress has been made in its understanding, so current knowledge on this topic has started to become relevant. In this context, recent findings have provided new insights, specifically into the correlations between the ferroptosis network and cancer, as well as between iron homeostasis, lipid metabolism and oxidative stress. These factors are now believed to be central players in tumorigenesis, tumor development, invasion, metastasis and therapy resistance. 

As a unique cell death process that is mechanistically and morphologically distinct from other forms of RCD, ferroptosis has attracted special attention from scholars and researchers who have immediately sensed the potential for new therapeutic opportunities in the modulation of this process. Indeed, over the past decade, drug-like pharmacological modulators targeting ferroptosis (both inducers and inhibitors) have been designed and developed at the preclinical level. In this context, growing evidence suggests the induction of ferroptosis as a promising strategy for the treatment of various types of cancer. Though the regulation of ferroptotic cell death seems highly heterogenic and strictly dependent on tumor phenotypes, its activation is generally regarded as a precursor to inducing cancer cell death, highlighting challenges in drug development towards efficient ferroptotic activity. From this perspective, recent clinical trials—in which the effects of molecules capable of acting on ferroptotic pathways have been evaluated alone or in association with chemotherapeutic drugs—are being explored with great interest (see Table 2). 

A further important contribution to the development of new anticancer strategies based on ferroptosis targeting could derive from CRISPR/CRISPR-associated nuclease 9 (Cas9). As well as being used to explore the genetic and molecular basis of cancer phenotypes, CRISPR/Cas9 technology is also used for the identification of potential molecular targets to suppress tumor growth and proliferation, allowing for significant progress in the treatment of specific tumors [204]. Applied to the study of ferroptosis as a defined RCD pathway to be exploited as a potential antitumor mechanism, this approach could effectively lead to significant advances grounded in the discovery of novel molecular targets for cancer treatment [205]. 

Accordingly, following various paths and different approaches, we are promptly advancing in knowledge for the identification of additional druggable targets relating to ferroptosis in order to provide novel therapeutic strategies for cancer therapy. Moving in this direction, forthcoming progress in selective modulators of ferroptosis can be envisioned. Alone or in combination with other antiproliferative agents, these modulators may improve the efficacy of antitumor treatments in clinics, especially for refractory and resistant cancer phenotypes.

## Figures and Tables

**Figure 1 cancers-16-01220-f001:**
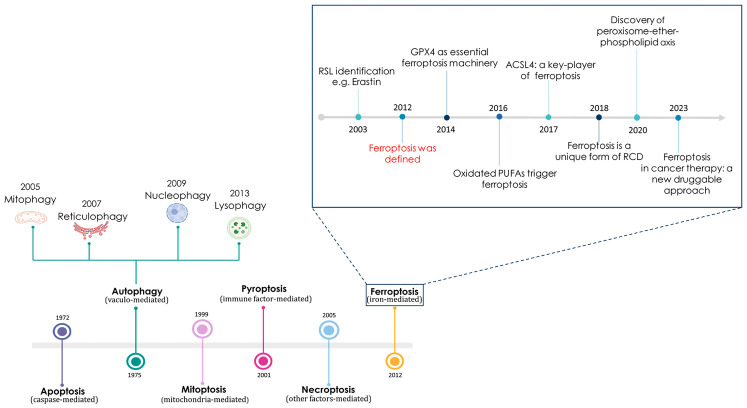
Timeline representing the discovery of the main regulated or unregulated cell death pathways, with a special focus on the essential findings concerning ferroptosis.

**Figure 2 cancers-16-01220-f002:**
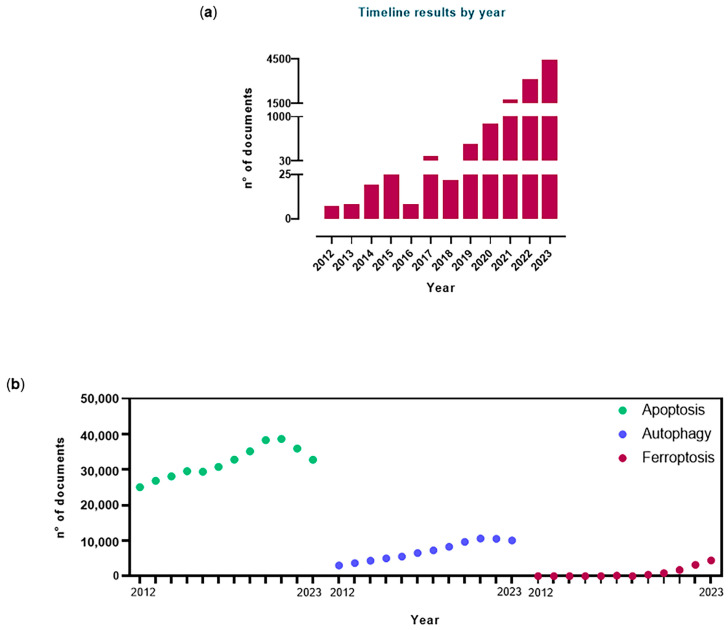
(**a**) Results by year over the last 12 years (2012–2023) of scientific studies pertaining to ferroptosis obtained through the PubMed database (accessed in February 2024); (**b**) the dot plot illustrates the search hits timeline from 2012 to 2023 correlated to scientific papers focused on apoptosis, autophagy and ferroptosis traceable in the PubMed database. The analysis of the scientific literature was performed considering both original research papers and reviews published in the PubMed database over the 2012–2023 time frame and featuring the words “apoptosis”, “autophagy” or “ferroptosis” in the title.

**Figure 3 cancers-16-01220-f003:**
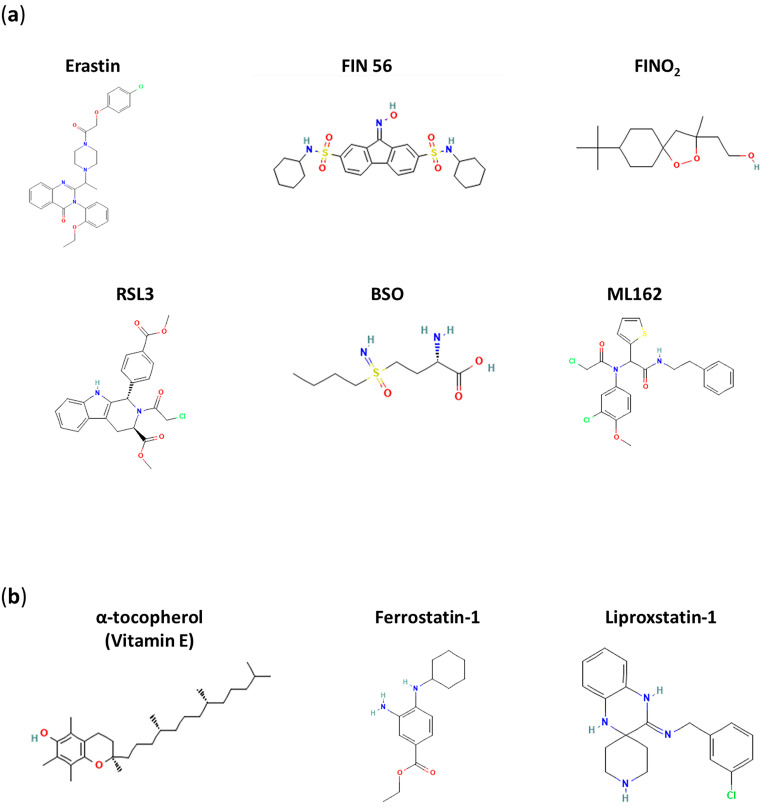
Molecular structures of some ferroptosis inducers (FINs) (**a**) and inhibitors (**b**). Heteroatoms in molecular structures are shown in color (nitrogen in blue, oxygen in red, sulfur in yellow, chlorine in green, hydrogen in light blue).

**Figure 4 cancers-16-01220-f004:**
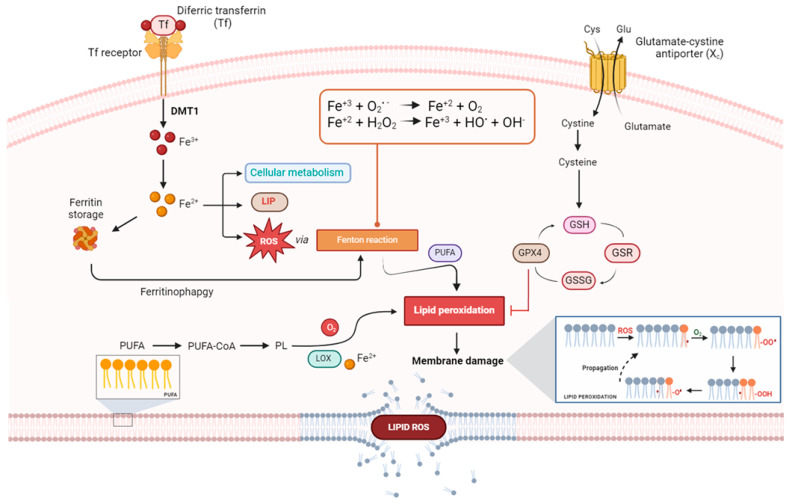
Schematic model proposed to describe molecular mechanisms underlying the ferroptosis cell death pathway. DMT1: divalent metal transporter 1; LIP: labile iron pool; ROS: reactive oxygen species; PUFA: polyunsaturated fatty acid; GPX4: glutathione peroxidase 4; GSH: reduced glutathione; GSR: glutathione disulfide reductase; GSSG: oxidized glutathione; LOX: lipoxygenases.

**Table 1 cancers-16-01220-t001:** Summary table reporting the molecular targets and mechanisms of action of the common inducers and inhibitors of ferroptosis.

Compounds	Inducer/Inhibitor	Molecular Targets	Mechanisms of Action
Erastin	Inducer	System Xc-	Cystine uptake inhibition, GSH depletion, LIP increase
L-buthionine solfoximine (BSO)	Inducer	System Xc-	GSH depletion
Sorafenib	Inducer	System Xc-	GSH depletion
Artesunate	Inducer	System Xc-	GSH depletion
RSL3	Inducer	GPX4	GSH activity inhibition
Altretamine	Inducer	GPX4	GSH activity inhibition
FIN56	Inducer	GPX4	GPX4 degradation, CoQ10 depletion
FINO2	Inducer	GPX4, Iron (II)	GPX4 inhibition, iron (II) oxidation
ML162	Inducer	GPX4	GSH activity inhibition
Ferrostatin (Fer-1)	Inhibitor	Lipid peroxidation	Lipid peroxidation inhibition
Liproxstatin (Lip-1)	Inhibitor	Lipid peroxidation	Antioxidant activity
α-tocopherol (Vitamin E)	Inhibitor	LOX	Lipid peroxidation inhibition

**Table 2 cancers-16-01220-t002:** References of some clinical trials encompassing ferroptosis.

Clinical Trial ID	Conditions	Treatment	Start Date	Phase
NCT06218524	Glioblastoma multiforme	Haloperidol, Temozolomide	December 2024	II
NCT06048367	Advanced solid tumors	[CNSI-Fe(II)]	October 2022	I
NCT04205357	Glioma, Glioblastoma	Sulfasalazine	March 2020	I
NCT03247088	Acute Myeloid Leukemia	Sorafenib, Busulfan, Cyclophosphamide, Fludarabine, Mycophenolate Mofetyil, Tacrolimus	July 2017	I/III
NCT02559778	Neuroblastoma	Sorafenib, Ceritinib, Dasatinib, Vorinostat, DFMO	September 2015	II

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
