# Peer review of "Insight into Iron, Oxidative Stress and Ferroptosis: Therapy Targets for Approaching Anticancer Strategies"

_cancers, 2024, doi:10.3390/cancers16061220_

Round 1

Reviewer 1 Report

Comments and Suggestions for Authors

Your manuscript entitled "Insight into iron, oxidative stress and ferroptosis: therapy tar- 2 gets for approaching anticancer strategies" provides a systematic review of iron, oxidative stress and ferroptosis that is commendable for its depth and breadth of coverage. The commitment to cover a wide range of aspects related to these critical biomedical topics is evident and highly appreciated. Specific comments for improvement include: 

1. Condensing the background information covered in previous reviews by Daolin Tang, Scott Dixon and Boyi Gan. Emphasising new insights rather than repeating familiar concepts. This will better serve the reader and the field between paragraph line 38 - 147.

2. Line 211, expand on the irreversible damage caused by iron overload with specific examples and mechanisms, such as how iron promotes lipid peroxidation and DNA damage. Providing deeper insights into the effects of iron overload on biomolecules and biological structures will enrich the manuscript.

3. Please explain how the cancer cell microenvironment, including hypoxia, nutrient deprivation, and intercellular communication, affects ferroptosis regulation and cancer treatment responsiveness.

4. The author would be good to explore the expression changes of TfR and FPN across various cancer types. how these alterations influence cancer progression and treatment? This examination could unveil potential therapeutic targets or diagnostic markers.

5. The manuscript discusses ferroptosis inducers such as erastin, RSL3, and FIN56. However, it would be more compelling to discuss their role in different cancer models and their interactions with the tumour microenvironment.

6. Investigate specific biomarkers for ferroptosis, such as those identified in PMID: 38401540. Proposing new markers or methods and discussing their mechanisms and controversial aspects could significantly contribute to the field.

7. Its good to discuss the potential for developing new anticancer drugs based on ferroptosis mechanisms, targeting strategies, and drug design ideas for specific cancer types, referencing PMID: 37716885 and PMID: 38297130.

8. Explain how the cancer cell microenvironment, including hypoxia with iron, nutrient deprivation links to iron, or intercellular communication, affects ferroptosis regulation and cancer treatment responsiveness. eg: studies such as PMID: 38351124 and PMID: 38366038.

9. author would be good to highlight the use of CRISPR/Cas9 and other gene editing technologies, methods, or new applications in ferroptosis research, with a focus on potential in vivo applications, offering a forward-looking perspective on future in vivo ferroptosis application.

10. Accurate citation of research data and results should be ensured while avoiding excessive citations of widely accepted concepts to streamline the text.

Comments on the Quality of English Language

The manuscript frequently revisits the concepts and mechanisms of iron in a manner that detracts from the conciseness and readability of the text. To refine your manuscript, I recommend focusing on key discoveries and novel findings, providing an overview of established knowledge while also highlighting recent advances or underexplored avenues of research. This balance will not only increase the value of the manuscript, but will also improve its clarity and impact.

Reviewer 2 Report

Comments and Suggestions for Authors

This manuscript presents the ferroptosis and oxidative stress leading diseases including cancers comprehensively. This manuscript is attracted to the readership of the Dalton Trans.; however, major revision of the manuscript is necessary prior to publication. The detailed comments are below.

1.      A summary figure of the mechanisms covering from ferroptosis and oxidative stress to cancer progression would be useful to understand the effects of ferroptosis in onset and progression of cancer. Although the text adequately describes the mechanisms, presenting them in a figure (or a table) would facilitate comprehension for readers.

2.      A summary table of ferroptosis inhibitors presenting their chemical structure, chemical properties and biological activity could be valuable.

Reviewer 3 Report

Comments and Suggestions for Authors

The submitted review focuses on the ferroptosis and related aspects. The review is very well written, from the scientific point of view I can call it an outstanding one. However I also recommend some revisions listed below.

Line 8, it is a bit unclear for me why the authors emphasize that only the studied within last 10 years have been considered in this review. In line 40, the authors state that the ferroptosis was first described in 2012, therefore 12 years ago. This 2 years gap (12 vs 10) is quite confusing.

Figure 1, while I really like this figure, the font size used is too small. It is impossible to read some words.

Figure 1, are there any RCD described after 2012?

Line 56, somewhere here it would be reasonable to include a figure presenting the molecular structure of erastin

In the whole manuscript the Authors use “iron” while they refer to iron ions, either iron(II) or iron(III). I know it may be difficult, but the authors should be precise and change, when possible, from “iron” to either “iron(II)” or “iron(III)”.

Figure 2b, what was the reason for repeating the x-scale three times? Can’t those three series be combined in one scale?

Line 117, here, the Authors should described, at least briefly, the mechanism behind the Fe(II)/Fe(III) balance regulation

Around line 250, since the authors have gone into that direction, it would be reasonable to mention the RDI and RDA for iron

Line 389, inappropriate style of reference

Lines 596-600, since this is a review and also due to the limited number of figures in this review, I recommend to create a figure presenting the molecular structures of the compounds described here

Reviewer 4 Report

Comments and Suggestions for Authors

A review by Piccolo et al "Insight into iron, oxidative stress and ferroptosis: therapy targets for approaching anticancer strategies" discusses the current knowledge on ferroptosis, a relatively novel type of regulated cell death. This review paper is very well written, and full of important pieces of information.

Specific comments:

Fig. 2 - please change "time(y)" to jusy "year" in both panels. In addition, in the panel B, all three types of cell death should be placed in a single X-axis (common for all types of cell death). The colors sufficiently differentiate them.

line 160 - a formula for superoxide should be written properly.

Table showing currently active clinical trials on drugs inducing ferroptosis in cancer would improve the quality of this manuscript.

Gene names should be written in italics.

Comments on the Quality of English Language

minor revision required

Reviewer 5 Report

Comments and Suggestions for Authors

Review article titled (Insight into iron, oxidative stress and ferroptosis: therapy targets for approaching anticancer strategies) discusses the therapeutic value of targeting iron and oxidative stress as a therapeutic target in cancer therapy. In fact, this is a useful review for the students studying in cancer field and biochemistry field

1- TItle is suitable and adequately represents the topic

2- Abstract : if possible, can it be amended by some numerical values?

3- Key words: suitable

4- Plagiarism % is high and it is recommended to be reduced to less than 10%

5- The resolution of figure 1 is poor, needs effeort to be enhanced

6- Give an introduction to give the rational and importance  of the topic 

7- Mention the methodology of search for resuts in this review

8- Describe how data in Figure 2 A and Figure 2B were achieved?

9- Item 4: better to be divided into 2 items (iron and cancer & OS and cancer)

10- Item 6: give separate subtitles: one for inducers and one for inhibitors

This section is too long and hard to be comprehended, please give subtitles as mentioned above and also divide them into relevant paragraphs

11- A graphic presentation for data in Item 6 will be a valuable addition to this review

12- Section 7 is also too long & needs to be divided into 2 parts

13- Section 8: divide it please into relevant paragraphs and sections:  it is very hard to read 3 pages as one shot!!

Section 9: please explore more the conclusion & separate paragraph for the future directions and the clinical potential of the review article as this point is very important

14- 

Comments on the Quality of English Language

fine

Round 2

Reviewer 2 Report

Comments and Suggestions for Authors

All my concerns were cleared. This revised manuscript is now suitable for publication.

Author Response

We thank the reviewer again for his time and work, and for the positive feedback

Reviewer 5 Report

Comments and Suggestions for Authors

The revised version of review titled (Insight into iron, oxidative stress and ferroptosis: therapy targets for approaching anticancer strategies.) is improved compared to the original version. but please consider the following

1- palgiairsm should be reduced 

2- A section for detailed methodology of the review must be added inside a text

Comments on the Quality of English Language

fine

Author Response

We thank the reviewer again for the precious advice that allowed us to improve this review.

- Plagiairsm should be reduced. 

We have really done everything possible to achieve this result (we believe the level of plagiarism is however limited) considering that this is a review and that many concepts are actually familiar and widely covered in the literature. In the effort we have made, the risk is that some concepts (especially in the more technical and scientific passages) may end up being altered in an attempt to completely reformulate sentences and texts.

- A section for detailed methodology of the review must be added inside a text

In line with the reviewer's suggestions, we have now added in the revised form of the manuscript a small section dedicated to the methodology.

Round 3

Reviewer 5 Report

Comments and Suggestions for Authors

Try again to reduce plagiarism to less than 10%

It is not impossible 

Comments on the Quality of English Language

Fine